# Etiology of Late-Onset Alzheimer’s Disease, Biomarker Efficacy, and the Role of Machine Learning in Stage Diagnosis

**DOI:** 10.3390/diagnostics14232640

**Published:** 2024-11-23

**Authors:** Manash Sarma, Subarna Chatterjee

**Affiliations:** Computer Science and Engineering, Faculty of Engineering and Technology, Technology Campus (Peenya Campus), Ramaiah University of Applied Sciences, Bengaluru 560058, India

**Keywords:** late-onset Alzheimer’s disease, biomarker, machine learning, deep learning

## Abstract

Late-onset Alzheimer’s disease (LOAD) is a subtype of dementia that manifests after the age of 65. It is characterized by progressive impairments in cognitive functions, behavioral changes, and learning difficulties. Given the progressive nature of the disease, early diagnosis is crucial. Early-onset Alzheimer’s disease (EOAD) is solely attributable to genetic factors, whereas LOAD has multiple contributing factors. A complex pathway mechanism involving multiple factors contributes to LOAD progression. Employing a systems biology approach, our analysis encompassed the genetic, epigenetic, metabolic, and environmental factors that modulate the molecular networks and pathways. These factors affect the brain’s structural integrity, functional capacity, and connectivity, ultimately leading to the manifestation of the disease. This study has aggregated diverse biomarkers associated with factors capable of altering the molecular networks and pathways that influence brain structure, functionality, and connectivity. These biomarkers serve as potential early indicators for AD diagnosis and are designated as early biomarkers. The other biomarker datasets associated with the brain structure, functionality, connectivity, and related parameters of an individual are broadly categorized as clinical-stage biomarkers. This study has compiled research papers on Alzheimer’s disease (AD) diagnosis utilizing machine learning (ML) methodologies from both categories of biomarker data, including the applications of ML techniques for AD diagnosis. The broad objectives of our study are research gap identification, assessment of biomarker efficacy, and the most effective or prevalent ML technology used in AD diagnosis. This paper examines the predominant use of deep learning (DL) and convolutional neural networks (CNNs) in Alzheimer’s disease (AD) diagnosis utilizing various types of biomarker data. Furthermore, this study has addressed the potential scope of using generative AI and the Synthetic Minority Oversampling Technique (SMOTE) for data augmentation.

## 1. Introduction

Early diagnosis of Alzheimer’s disease can be beneficial for precluding its progression. “Accumulation of the protein beta-amyloid outside neurons and twisted strands of the protein tau inside neurons are hallmarks. They are accompanied by the death of neurons and damage to brain tissue. Inflammation and atrophy of brain tissue are other changes” [1]. A disruption in neuronal connectivity is observed within the brain. The hippocampus appears to be the region of origin for this neurological disturbance. As the neurons die, other parts of the brain are subsequently affected. This process initially manifests as impairment in short-term memory formation. Subsequently, there is a progressive deterioration in short-term memory function. This is followed by a decline in other cognitive faculties and the emergence of behavioral issues [2]. This study focuses on late-onset AD which typically manifests after the age of 65. While early-onset AD primarily results from genetic factors, multiple etiological factors contribute to the development of LOAD [3]. It has been determined that 50–75% of people aged 65 years or above are prone to Alzheimer’s disease. As global life expectancy increases, the number of Alzheimer’s patient is also concomitantly rising worldwide.

To date, no major cure for this disease has been established [4]; clinical trials for AD drugs have demonstrated a failure rate of 99.6% [5]. On 7 June 2021, the Food and Drug Administration (FDA) sanctioned the first novel medication for Alzheimer’s disease in several decades. However, the euphoria was clouded by apprehensions regarding unknown efficacy, considerable adverse effects, a wide target patient demographic, elevated prices, and a possible correlation with heightened demand for diagnostics [6]. Currently, several pharmaceutical research initiatives are underway to slow the progression of the disease or delay the cognitive decline [7]. Still, we are far from finding a definitive solution or cure. In this scenario, early diagnosis of the disease stage appears to be the only viable option. Such an approach would facilitate the minimization of progressive deterioration through preventive measures. This prompts crucial research inquiries: What are the factors contributing to AD? Can these risk factors be identified at an earlier stage? Can AD be diagnosed early? What are the early biomarkers of AD? Can all AD biomarkers be utilized in routine clinical diagnosis? Can ML techniques be effectively employed in early AD diagnosis with high accuracy? How effective are ML techniques in terms of AD diagnosis? The objective of this study is to find answers to these questions. Consequently, this study addresses the cause, analysis, and diagnosis of Alzheimer’s disease (AD) utilizing diverse biomarker data and evaluating their efficacy through ML techniques. Some surveys have already been conducted on early diagnosis of AD using AI and DL techniques. Vrahatis et al. conducted a research survey on the early detection of Alzheimer’s disease utilizing non-invasive biomarkers through the application of artificial intelligence (AI) and deep learning techniques [8]. Although their study focused on the importance of AD detection utilizing non-invasive and sensor-based biomarkers through AI and DL techniques, there was no significant discussion regarding the diverse risk factors associated with AD that are reflected in different genetic and other early biomarkers. Our study on LOAD detection is conducted from a 360-degree angle. This study incorporates a three-pronged outcome strategy that can be beneficial not only for researchers but for all stakeholders involved in AD diagnosis.

The advent of machine learning (ML) technology has revolutionized the early detection of disease stages through the analysis of biomarker datasets. ML is now extensively utilized in disease diagnosis due to the increasing availability of AD datasets. An expert’s ability to diagnose a condition from its biomarkers has been heavily reliant on their clinical experience. However, this conventional approach is often time consuming and susceptible to human error. Thus, the application of machine learning-based models for disease diagnosis, including AD, is gaining popularity in recent years. However, this approach presents several challenges. One notable challenge is the prevalence of incomplete and unstructured data within the biomarker set. The process of identifying and selecting pertinent biomarkers from an extensive set poses a different challenge. Moreover, there are various other challenges to consider. LOAD exhibits multiple etiological factors and pathways leading to its inception. The complexity of mechanisms involved in LOAD presents significant challenges in developing an effective approach for early prediction based on specific data.

Depending on the etiological factors contributing to LOAD and the stage of disease progression, AD may be reflected in different biomarker samples extracted from an individual. Genetic biomarkers, for instance, can reflect the risk of AD well in advance of symptom onset in an individual and serve as valuable indicators for preclinical analysis. Conversely, age and cerebrospinal fluid (CSF) are utilized in clinical analysis of the disease. This study has explored the etiological factors of LOAD, the biomarkers employed in the clinical and preclinical diagnosis of AD, and the current research on diagnosing AD stages through ML techniques utilizing various biomarkers. The primary objective of this paper is to investigate the factors contributing to LOAD, as well as its symptoms and the associated biomarkers at both clinical and preclinical stages of the disease, and to analyze the existing literature on the application of ML techniques for diagnosis. Given that the literature identifies the DL family of techniques as the most prevalent one, we have discussed the potential applications of DL techniques in AD diagnosis. This research aims to provide a multifaceted overview of the subject to assist future researchers in this field. 

## 2. Disease Cause and Biomarkers

### 2.1. Factors Leading to LOAD and Its Diagnosis

AD is primarily defined as a decline in cognitive function and memory. “The two hallmark pathologies required for a diagnosis of Alzheimer’s disease (AD) are the extracellular plaque deposits of the β-amyloid peptide (Aβ) and the flame-shaped neurofibrillary tangles of the microtubule binding protein tau.” [9]. So, the “gold standard” of AD diagnosis is the presence of amyloid plaques and neurofibrillary tangles in the brain. However, the pathological processes of AD commence decades prior to the manifestation of such clinical features.

Figure 1 illustrates the ‘System Biology Approach’ block, which comprises two components: ‘molecular networks/pathways’ and ‘healthy vs. disordered brains’. These components are encompassed by blocks representing multi-omics data, including genomic, transcriptomic, epigenetic, environmental, and metabolic factors. The multi-omics data potentially influence the ‘molecular networks/pathways’, which may subsequently contribute to the development of healthy or disordered brains, as illustrated in the second component of the block. These multi-omics datasets represent the factors that cause AD. A study of these factors may facilitate the early detection of AD risk. We consider these datasets as potential early biomarkers. Additional multi-omics data, such as structural and functional brain imaging and the connectome, reflect the brain structure, connectivity, and functionality in both healthy and disordered brains, which constitute the second component of the ‘System Biology Approach’. This second component represents the impact of the first component (molecular networks/pathways). The brain structure, connectivity, functionality, etc. collectively shape an individual’s behavioral patterns. In the clinical context, these multi-omics data serve as a valuable set of biomarkers for AD diagnosis and are considered as biomarkers at the clinical stage.

### 2.2. Alzheimer’s Disease Biomarkers

Biomarkers can be categorized according to their utility in the diagnosis and prognosis of disease. As per National Institute of Health in the year 2001, “Biomarker is an indicator of certain objective measure and evaluation of biological process, pathogenic process, or pharmacological evaluation of therapeutic efficacy”. As per the 1998 Report of the Ronald and Nancy Reagan Research Institute of the Alzheimer’s Association and the National Institute on Aging Working Group titled “Molecular and Biochemical Markers of Alzheimer’s Disease”, an AD biomarker ideally has sensitivity and specificity greater than 80%, while a positive predictive value needs to be above 90% for detecting AD. Given the multifaceted etiology of Alzheimer’s disease (AD) (as illustrated in Figure 1) and the current clinical, neurophysiological state, brain structure, and functional state utilized in AD diagnosis, AD biomarkers (Figure 2) can be categorized according to their application in clinical and preclinical trials. Biomarkers used for the diagnosis of AD in clinical trials—including the identification of disease risk at an early stage—encompass cognitive assessments such as the mini-mental state examination (MMSE), neuroimaging biomarkers such as magnetic resonance imaging (MRI), cerebrospinal fluid (CSF), diffusion tensor imaging (DTI), and blood plasma AD biomarkers. Genome sequencing, gene/RNA expression profile, protein expression profile, epigenomic expression for deoxyribonucleic acid (DNA) methylation, gene–gene interactome, and gene–protein interaction data derived from an individual’s profile can potentially detect Alzheimer’s disease (AD) risk at the preclinical stage and may be considered preclinical biomarkers.

The biomarkers used in preclinical AD diagnosis can detect genetic mutations, gene expression patterns, and predict significantly in advance the accumulation of β-amyloid (Aβ) peptides, neurofibrillary tangles (NFTs), and synaptic dysfunction at a later stage in the patient’s brain, leading to Alzheimer’s. The Genome Wide Association Study (GWAS) is able to detect single-nucleotide polymorphisms or SNP variations in the genomes of patients with AD. “Although there is evidence of an important genetic component in AD, the majority of AD is probably caused by complex interactions between one or more susceptibility genes and different environmental factors” [11]. Epigenetics is the study of the impact of behavior and environment on gene function. This mechanism is an important regulator of gene expression that ultimately affects the formation of different proteins, including those causing AD. Genomic, epigenetic, and other early biomarkers are primarily useful in AD diagnosis in the pre-clinical stage of the disease long before the disease symptoms start manifesting. In contrast, biomarkers such as MMSE, blood plasma, PET scan speech, text, etc., are valuable when symptoms begin to manifest. Biomarkers utilized in clinical AD diagnosis reflect the current neuropsychological state and brain structure/function, and they are efficacious for the classification of AD stages—cognitive normal (CN), mild cognitive impairments (MCI), and Alzheimer’s disease (AD). Cognitive tests, patient age, MRIs for evaluating brain structural and functional integrity, invasive CSF biomarkers for quantifying tau levels and amyloid, fluorodeoxyglucose positron emission tomography (FDG PET) region of interest (ROI) averages for assessing metabolism in cells, AV45 PET ROI averages for measuring amyloid-beta load in the brain, and DTI ROI for analyzing microstructural parameters of axons and other biomarkers (such as APOE status, demographic information, diagnosis) are predominantly utilized in clinical practice to identify the stage of Alzheimer’s disease. The APOE status with increasing APOE4 allele count increases the risk of late-onset Alzheimer [12]. Research has demonstrated that the speech or utterances of subjects can aid in differentiating between cognitively normal individuals and Alzheimer’s disease (AD) patients by utilizing various audio and text features, including linguistic inquiry and word count (LIWC), word vectors, bidirectional encoder representations from transformer (BERT) embeddings, and computerized language analysis (CLAN) features. These features can be considered as biomarkers at the clinical stage.

The manual analysis of biomarkers in AD diagnosis is time consuming and prone to human error. ML-based data-driven disease diagnosis is progressively being adopted by the health industry. This review encompasses research papers on AD detection utilizing ML-based techniques using biomarker data. The strategy and purpose of the review are illustrated in Figure 3. We collected data on both early AD biomarkers and those reflecting the effects of AD, such as brain functionality, structure and an individual’s behavior, which are valuable in clinical stage diagnosis. This study selected specific biomarkers from each category and subsequently analyzed research papers that focused on AD diagnosis utilizing the chosen biomarker data. The intended outcomes at the broad level of our study encompass the identification of research gaps, assessment of biomarker efficacy, and determination of the most effective or prevalent ML technology utilized in research.

## 3. Literature on Alzheimer’s Disease

Numerous studies have already been conducted on AD diagnosis using ML techniques from various biomarker data encompassing demographic, socioeconomic, clinical, cognitive, imagery, genetic, epigenetic, and linguistic performance measures. A substantial body of research literature has been published on this subject. We have examined several of these publications. To attain the intended outcome of our study with a focus on the clinical and preclinical aspects of diagnosis, we selected specific research papers for review. Of these, six research papers belong to the ML-based AD diagnosis from the imagery category of MRI and PET scan images. Three papers belong to the category of genetic and epigenetic data-based diagnoses. Another two belong to diagnosis based on clinical, demographic, and non-imagery biomarker data. Four research papers belong to the ML-based diagnosis from linguistic biomarkers. This study covers three research papers on AD diagnosis based on the integration of different biomarker types and multimodal analyses. We cover a research paper on attention-based image classification. We separately studied three research and review papers on biomarker data, including invasive and noninvasive types. Table 1 presents a selection of few research papers on AD diagnosis using ML methods followed by detailed exposition.

Zhang et al. had proposed a classifier of the “sparse representation type” (SRC) [22] with an intention to create local patch based sub-classifiers, which is an ensemble method for MRI image samples from 652 subjects of Alzheimer’s Disease Neuroimaging Initiative (ADNI). First, they partitioned each image into several local patches. Then, a subset of patches is randomly selected to build a weak classifier, and the ensemble technique is applied to the subclassifiers. Thus, they could increase the sample size, capture local spatial features, and increase the model performance with an accuracy of 90.8% and an area under the ROC curve (AUC) of 94.86% for AD classification and an accuracy of 87.85% and an AUC of 92.90% for MCI classification. Westman et al. conducted research with CSF and a baseline MRI biomarker combination to enhance accuracy [13] compared with when each biomarker was used individually. Their dataset included 96 samples from 273 CN and 96 AD patients. Their proposed classification had 91.8% accuracy with a combination of CSF and MRI. Veeramuthu et al. used PET brain images to classify subjects with AD and CN. They first performed feature extraction using spatial normalization and noise filters [23]. They applied the Fisher discriminant ratio for feature extraction to obtain the region of interest (ROI) and trained them with the associative rule (AR) mining algorithm to achieve 91.33% accuracy, 100% specificity, and 82.67% sensitivity. However, they did not address the handling of missing data, management of data with an imbalanced class, or the validation process. Fulton et al. designed a gradient boosted machine (GBM) to predict AD from cognitive and sociodemographic data with 91.3% accuracy. Their ResNet-50 framework based on DL and CNN predicted a multiclassification accuracy of 98.99% from MRI [14]. Nevertheless, due to the limited sample size of only 416 individuals in the study, the findings cannot be generalized. Lella et al. designed an ML framework for classifying AD and analyzing feature importance from DICOM images acquired from the ADNI database [24]. Artificial neural network (ANN), support vector machine (SVM), and random forest (RF) techniques were applied to classify the information content of the communicability metric of the image samples collected from AD and CN natives. They achieved a connectivity matrix representing the structural complexity of the brain network from each subject by applying image processing techniques. Their ANN model achieved an AUC score of 83% and an accuracy of 75% in classification. Although the performance score was competitive, they skipped the MCI stage of classification. Sarma et al. [25] applied different machine learning techniques to predict AD stages (CN, MCI, and AD) and achieved the best F1 scores of 89% for CN, 84% for MCI, and 80% for AD stage identification using deep learning from ADNI baseline clinical study samples from 2000 subjects. They applied Sequential Floating Backward Selection (SFBS) and correlation matrix techniques to reduce dimensionality from 113 biomarkers to 8 biomarkers. However, DL techniques are generally sensitive to the nature of training data. They are also sensitive to random initialization, which is often overlooked and has not been addressed.

Bae et al. [26] in their research identified AD from T1-weighted MRI images of the medial temporal lobe. Inception-v4, a 2D image classification CNN pretrained model, was used. They used two datasets from ADNI and the Seoul National University Bundang Hospital (SNUBH), trained the fine-tuned model with 156 AD patients and 156 CN controls from each dataset, and tested the final model with 39 AD patients and 39 CN from each dataset. From each dataset, five model instances were constructed using 5-fold cross-validation. The average ensemble values of the average probabilities of the models generated from cross-validation were used as the final results in the test. They obtained an AUC score of 0.94 for the ADNI-trained model with ADNI test data and an AUC score of 0.91 for the SNUBH-trained model with SNUBH test data. For the ADNI-trained–SNUBH test and SNUBH-trained–ADNI test data, the AUC scores were 0.88 and 0.89, respectively. Fathi et al. [20] selected six of the best individual CNN-based classifiers to combine and construct an ensemble model for classifying AD stages. They achieved accuracy rates of 98.57, 96.37, 94.22, 99.83, 93.88, and 93.92 for NC/AD, NC/EMCI, EMCI/LMCI, and LMCI/AD, four-way and three-way classification groups, respectively, from the ADNI MRI dataset. The ensembling technique increased the performance far better than the individual model performance; however, when tested with a local MRI dataset, the performance was poor. In addition, they used accuracy as a performance measure for multiclassification, which is not recommended.

Huang et al. [27] developed an SVM-based method to classify AD genes from gene network data of human brain and gene expression in the whole genome. Candidate genes of AD were classified with an accuracy and ROC of 84.56% and 94%, respectively. This methodology provides a complementary approach for the spectrum of AD-associated genes identified from more than 20,000 genes on a genome-wide scale. Lee et al. in their study identified AD-related genes from DEGs (Differentially Expressed Genes), the TF (Transcription Factor) database, gene connectivity network data, and CFG (Convergent Functional Genomics) from blood gene expression data [16]. The AD related gene expression data are used to construct ML models from logistic regression (LR), L1-regularized LR (L1-LR), SVM, RF, and DNN models. The best average values of the AUC (area under the curve) obtained were 0.657 for ADNI, 0.874 for ANMI, and 0.804 for the ANM2 dataset with five-fold cross-validation for each dataset. These results suggest that gene expression data from blood samples are useful for predicting the AD stages. However, high data imbalance in the ADNI data where the minority dementia sample size is very small compared with the other two categories leads to a poor AUC score in the case of ADNI. Additionally, multiclassification is avoided here, even though the datasets have three labels, including the MCI stage. Park et al. in their research [17] proposed a DL-based model that can classify AD using integrated large-scale DNA methylation and gene expression data to construct multi-omics datasets. They achieved an accuracy of 0.823, which was better than that of a single omics dataset. However, they did not use the multi-omics dataset from the same sample group.

Orimaye et al. [28] used features like syntax, n-gram, etc. to build AD diagnostic models using SVM-based ML algorithms. It learns linguistic biomarkers of verbal sounds in elderly individuals, which can help in the clinical diagnosis of AD. The linguistic features were transcribed speech from 99 control individuals and 99 patients with AD from the Dementia Bank dataset. However, the small size of the datasets is the main limitation. Eyigoz et al. [19] used a logistic regression classifier to predict AD at a later stage from linguistic and non-linguistic biomarker samples collected when subjects were cognitively normal. They found that models based on linguistic variables performed better than predictive models that incorporated APOE, demographic variables, and NP test results from 703 samples from 270 Framingham Heart Study participants. The aggregation of both produced better results with the AUC. Haulcy et al. [20] classified AD from non-AD and MMSE prediction based on audio and text data from Alzheimer’s Dementia Recognition through Spontaneous Speech (ADReSS) dataset. They trained five classifiers using text and audio features. The SVM classifier that is trained on BERT embedding and CLAN feature combination with principal component analysis (PCA) dimensionality reduction technique performed the best. The average accuracy is 0.898. However, the accuracy of the speech data is poor.

Golovanevsky et al. developed a multimodal attention-based deep learning model for multiclass classification support in AD diagnosis [15]. The study utilized a cohort of 239 patients for whom data were available from all three modalities—imaging (551 patients), SNP (805 patients), and clinical (2284 patients). Using this multimodal combination, they obtained an F1 score of 91.41%. However, the performance significantly decreased when any of the modalities were removed, especially when clinical data were withheld. This again establishes the effectiveness of multimodal and clinical data.

## 4. Result of Literature Study

### 4.1. Research Gap Analysis

Several research gaps have been identified in previous research. We categorized these gaps at a broad level.

First, inherent issues are prevalent in the dataset itself, such as a low sample size and imbalanced data, which are common in the medical and healthcare domains. Numerous prior investigations were constrained by limited datasets comprising fewer than 500 samples due to data scarcity. Although cross-validation methodologies are beneficial in model development from small datasets, facilitating the capture of data variability, it has been noted that a significant portion of earlier research did not employ these techniques. The researchers employed the random splitting of training and test data for model training and validation. Consequently, while the performance results may appear impressive, their reliability is questionable.

Second, there is a paucity of research on intermediate stages of AD. Some recent research has impressive results with the binary classification of disease states AD and CN. It is observed that researchers preferred to use binary classification by considering the intermediate stage MCI as CN. While this approach may have yielded improved results, it potentially overlooks a crucial aspect of research by disregarding the intermediate stages of the disease. One reason may be that some AD datasets do not include intermediate stages such as the MCI stage, and it appears that AD identification has been the primary goal. However, the MCI stage is a preclinical stage that can significantly indicate the progression of AD.

The third gap concerns the limitations of GWAS-based AD research. Some researchers focused on GWAS for AD detection by identifying SNP alone. Thus, some genes, such as APOE4 and APP, can be identified as AD risk factors. However, AD genes such as APOE4 do not always cause AD. For AD with complex genetic mechanisms and diverse pathways involved, it is necessary to consider additional factors such as gene expression regulation and epigenetic expression in an integrated manner.

Next, it is about handling data of high dimension and low sample size (HDLSS) for AD diagnosis. The current research uses gene expression data for AD diagnosis. However, the diagnostic performance of these data is often low. A potential explanation for the suboptimal performance is that gene expression data exhibit HDLSS characteristics and, in some instances, are imbalanced, as exemplified by the ADNI gene expression dataset. While inherent complexity is involved with the data, better feature selection and classification techniques coupled with data augmentation would have increased classification performance. Furthermore, researchers did not employ proper performance matrices for AD classification measurement from imbalanced data, which is very common in real-world datasets. They used accuracy and AUC as performance measurements and omitted the F1 score. The F1 score is more accurate in measuring the performance of a model with all types of test data, particularly for imbalanced data.

### 4.2. Recommended ML Methods

ML-based AD diagnosis follows basic steps—feature selection/feature reduction and the classification of AD stage of a native from any AD biomarker dataset, which is followed by the performance evaluation of classification. Nonetheless, the choice of suitable algorithms for feature selection, dimensionality reduction, and classification depends on the particular AD biomarker collection, sample size, and dimensionality, as evidenced by prior research. Based on our comprehensive review of the AD literature, we have compiled a list of widely recognized and efficacious methodologies at a broad level for AD detection and classification of disease stages in Table 2. Nevertheless, some useful approaches of early research can be substituted by new methods and software libraries following thorough examination. For instance, early research utilized image slicing techniques for local patch selection during feature extraction. However, with the advent of CNN-based filters, CNN has emerged as a superior alternative now.

A common issue encountered during the construction of a classification model is the insufficiency of samples and data imbalance. One approach to augment the sample size is to amalgamate datasets with comparable column fields. Two NCBI gene expression datasets were merged to increase the sample size before the feature selection process [17]. Furthermore, given the availability of numerous efficacious pretrained models, researchers may not find it necessary to collect all samples and construct a model from scratch.

Emerging AIML technologies present a potential area for utilization. The advent of generative AI and the Synthetic Minority Oversampling Technique (SMOTE) has facilitated the application of data augmentation in model training scenarios characterized by limited sample sizes and significantly underrepresented minority samples. Synthetic data preserve all the underlying patterns and behaviors of the original dataset while altering the actual content [29]. ‘Gretel’ and ‘MOSTLY AI’ are two popular platforms for synthetic data generation through generative AI. Nevertheless, when utilizing synthetic data, it is imperative that the generated data be employed exclusively for training purposes, whereas validation or holdout data should comprise the original non-synthetic data [30]. Large Language Models (LLMs) can effectively facilitate the differentiation of text/transcripts between patients with Alzheimer’s disease (AD) and cognitively normal individuals.

It is observed that the DL family of techniques is widely used, although they require more extensive data for training. With the increasing availability of datasets through both natural and synthetic ways, the DL family of algorithms is gaining prominence.

Transformer and attention-based LLM techniques have been recently adopted for image classification. Mohanty et al. [31] developed an attention-based image classifier that accurately detects recurring odontogenic keratocysts (OKCs) from whole-slide images. It uses state-of-the-art transformer architecture with long-term and short-term memory. This has helped detect essential features and assisted pathologists to focus on detecting recurring OKC. The proposed model has a high classification accuracy of 0.98 and a recall of 1.0. This research suggests that methods other than CNN can be explored for image classification. This attention-based approach can be explored in AD diagnosis using scanned image data from the brains of natives.

### 4.3. Efficacy of AD Biomarkers

This study yielded insights into biomarkers, their utilization, and efficacy. It was observed that individual biomarkers produce low-to-moderate performance. However, when combined, they produce better results. This is attributed to the multifactorial etiology of AD, specifically LOAD.

Studies have shown that diagnostic performance results obtained from gene expression and gene sequence biomarkers are comparatively lower than those of MRI, CSF, and other clinical biomarkers. The primary reason for this is that many pathways are involved before AD-specific protein creation takes place from genes. Gene sequences exhibiting SNPs may indicate genetic mutations or anomalies. However, at an early stage, these may not necessarily translate into AD-specific risks due to the involvement of numerous other factors. Furthermore, gene expression data are generally of high dimension and small sample size (HDLSS). Nevertheless, these biomarkers can be utilized in diagnosis at the preclinical stage, significantly prior to the manifestation of AD symptoms. This can provide ample time for early treatment of the disease. The results show that biomarker combinations used in clinical trials, such as CSF, MRI, DTI, and MMSE, generally provide better diagnostic performance.

The collection of biomarker samples can be characterized as either simple or complex, depending upon the specific type of biomarker. The collection process may be classified as invasive, non-invasive, or minimally invasive. For instance, obtaining a CSF sample necessitates a procedure known as a spinal tap, which is alternatively referred to as a lumbar puncture. In contrast, the collection of speech, text, and blood gene expression samples involves non-invasive or minimally invasive procedures that are relatively simple and cost-effective [7,8,32]. Consequently, CSF biomarkers are not recommended for routine AD diagnosis in clinical settings but are useful for research purposes [33,34]. The information pertaining to AD biomarkers has been incorporated into Table 3.

## 5. Deep Learning in AD Diagnosis

Our study suggests that the DL family of techniques frequently produces superior outcomes and is being increasingly adopted by the ML research community for AD diagnosis. Genomics and related fields generate substantial volumes of data utilized in experimental research. For instance, a single human genome sequence comprises more than six billion bases. Deep learning is particularly well suited for processing this type of data.

Neural networks such as deep learning and CNN (Convolution Neural Network) in general can be utilized for AD classification, pattern analysis, dimensionality reduction in neuroimaging, gene expression, DNA sequencing, transcription factor (TF) binding, and sound and text data processing (Figure 4). Deep learning-based autoencoders can be used to transform high-dimensional data (such as genomic and image data) to low-dimensional data, and an appropriate classifier can subsequently be used to identify the AD stage from low-dimensional data [35].

CNN is already in use for two-dimensional image classification and object recognition. PET and MRI image samples from patients can be used as inputs to train the AD classifier and subsequently detect the stages of AD. CNN can be used to predict SNP sites in a single dimensional DNA sequence (as illustrated in Figure 4A) and consequently identify AD risk genes/gene alleys [36]. 

Regulatory regions in linear DNA sequences and DNA/RNA binding sites with proteins (TF) can be identified and predicted utilizing CNN. TF-binding sites influence the expression of target genes. CNN models are already in use to classify emotions and identify tones from speech and text data. Consequently, these can be used to predict verbal symptoms. Before high-dimensional AD biomarkers are passed to the AD classifier for stage detection, dimensionality reduction in data is a major step. Feature extraction and selection can be conducted utilizing the convolutional and pooling layers of a CNN. The resultant reduced biomarker dataset can subsequently be passed to the classifier for training and prediction of the AD stage. The classifier at the end is a fully connected deep classifier.

Deep learning models demonstrate the capacity for flexibility and scalability in proportion to the volume of training data. However, they exhibit sensitivity to the characteristics of the training data. Additionally, these models are susceptible to the effects of random initialization, as they employ a stochastic training algorithm for learning. Therefore, a different set of weights may exist for each training time, resulting in different predictions. One solution to such a high variance is to train multiple DL models and combine their individual predictions. This helps reduce prediction variance and improvise performance as well. An autoencoder (AE) based on deep learning can be used to reduce the dimensionality of data prior to transmission to the classifier. Employing an unsupervised approach, a standard AE extracts a low-dimensional representation of the input data.

CNN is effective for processing matrix data; however, many biomarkers are based on network connectivity, which is an area where CNN exhibits limitations. Graph convolutional neural network (GCNN) is an emerging technology based on CNN. In the context of a complex neurodegenerative disease such as AD, which is characterized by synaptic dysfunction, the analysis of irregular graph-structured data is crucial. In this regard, GCNN demonstrates potential as an effective analytical tool.

## 6. Summary and Conclusions

With the objective of understanding and disseminating information regarding the early diagnosis of LOAD, we conducted a comprehensive yet concise study. This study encompassed an analysis of LOAD’s etiological factors, its neurological impact, associated behavioral changes, their representation in diverse biomarker samples, and the application of machine learning methodologies in diagnostic processes. Our study of AD literature encompassed ML-based AD diagnoses derived from genetic, epigenetic, imaging, and linguistic data, including hybrid and multi-omics datasets. The outcomes of this study were categorized into three main areas: identification of research gaps, recommendation of effective machine learning techniques, and evaluation of the efficacy of various biomarker types in both the clinical and preclinical stages of the disease. Our findings demonstrate that DL and CNN techniques have been effectively utilized in the majority of the research. We have elucidated the application of DL and CNN techniques for LOAD diagnosis using diverse types of biomarker data. 

The identification of gaps in previous and current research, as one of the outcomes of this study, and the addressing of some of these gaps using recommended ML techniques (another outcome), are anticipated to benefit future researchers. This study has identified several emerging technologies, notably generative artificial intelligence (gen AI) and synthetic data generation (SDG), which show promise in diagnostic applications and offer potential solutions to specific challenges. The evaluation of diverse biomarkers for Alzheimer’s disease (AD) diagnosis, encompassing their effectiveness in both preclinical and clinical stages, the invasive nature of sample acquisition, and their diagnostic accuracy, can inform research priorities and guide the implementation of these markers in clinical practice and routine AD assessment. Although our study addresses all aspects of LOAD from a multifaceted perspective, we were unable to delve into every specific detail due to the extensive nature of LOAD research and the multitude of factors involved. One limitation of our study was the inability to focus on AD progression, which constitutes another significant area of AD research. Furthermore, this study did not encompass research on AD risk factors such as environmental conditions, aging, and lifestyle habits, which are significant concerns in contemporary society. These aspects remain as potential avenues for subsequent studies.

This study aims to benefit diverse range of stakeholders, encompassing researchers, healthcare practitioners, clinical specialists, and other relevant parties. It is anticipated to facilitate their rapid and comprehensive understanding of the multifaceted aspects associated with LOAD, which involves intricate mechanisms. Furthermore, this study aims to empower these stakeholders to adopt suitable methodologies incorporating machine learning-based techniques for the timely identification of the disorder.

## Figures and Tables

**Figure 1 diagnostics-14-02640-f001:**
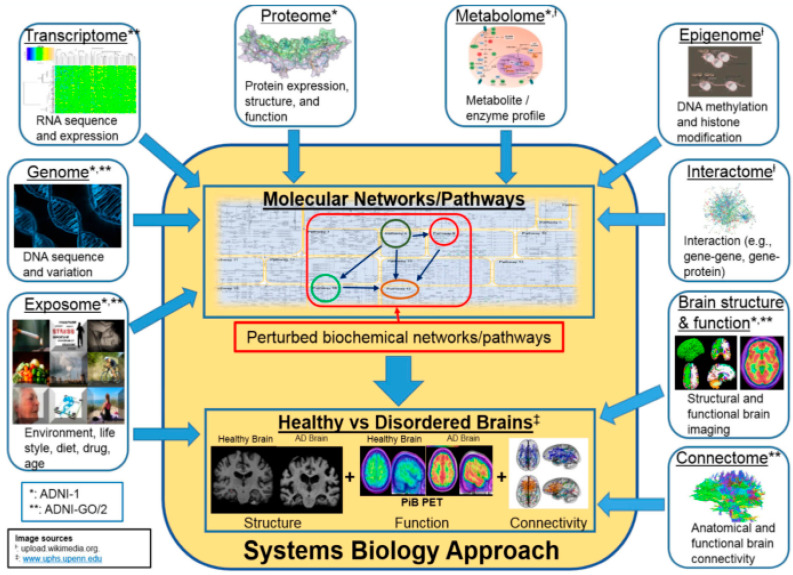
AD diagnosis—a system biology approach, source (s)—Wikimedia Com and review paper [10].

**Figure 2 diagnostics-14-02640-f002:**
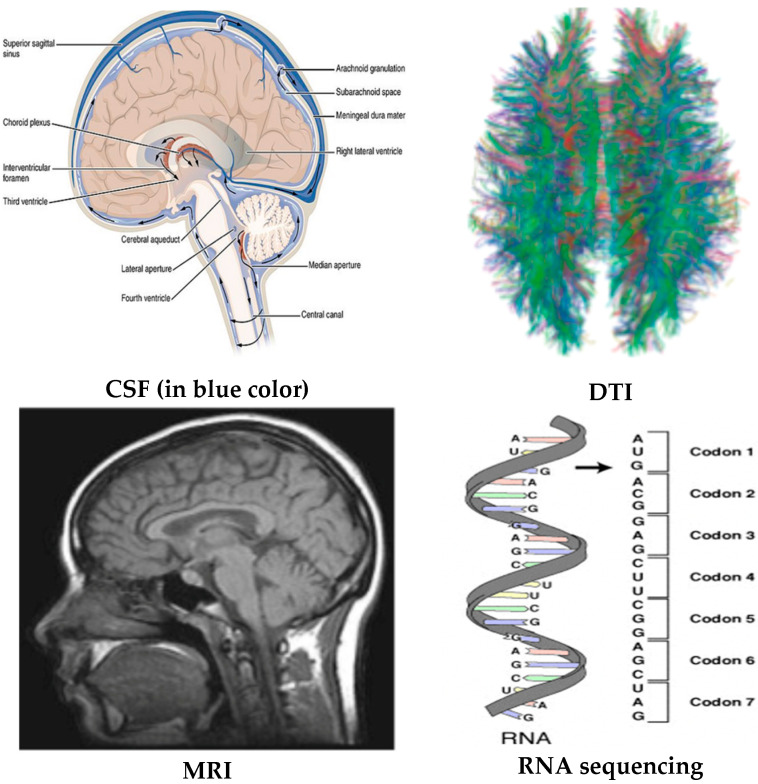
AD biomarkers, source (s)—Wikimedia Commons.

**Figure 3 diagnostics-14-02640-f003:**
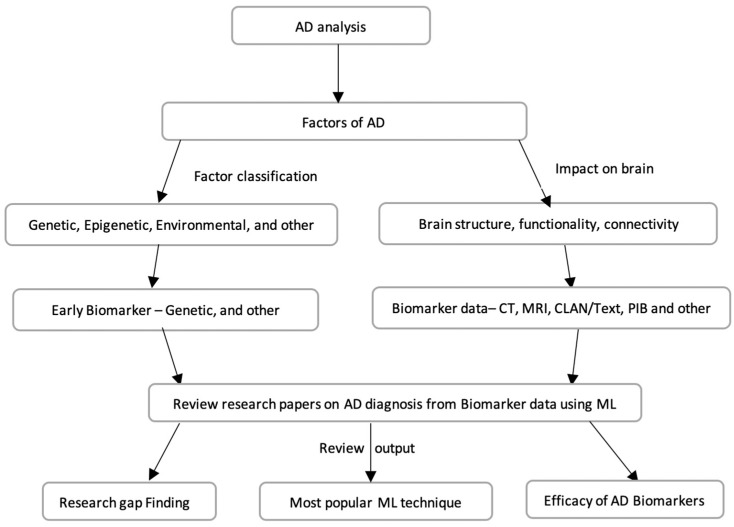
AD literature review strategy and outcome.

**Figure 4 diagnostics-14-02640-f004:**
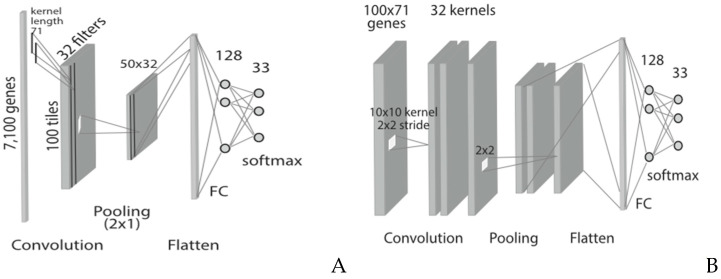
Different CNN architecture diagrams—(**A**): CNN with single dimensional input applicable for text, genomic data, (**B**): CNN with two-dimensional input. (**A**,**B**)—courtesy Mostavi et al. [36].

**Table 1 diagnostics-14-02640-t001:** AD literature study highlight.

Study	Biomarker Dataset (s)	Feature Selection	Classifier	Results
Westman et al. (2012) [13]	Clinical CSF and baseline MRI data combination	Did not use	Orthogonal partial least squares (OPLS) tool	81.6% for CSF and 87.0% for MRI separately 91.8% accuracies together
Fulton et al. (2019) [14]	ADNI clinical study data and MRI image	GBM-based feature importance	GBM, ResNet-50	91.3% prediction accuracy with GBM, 98.99% for image with ResNet-50
Golovanevsky et al. (2022) [15]	ADNI clinical, MRI and SNP based genetic	Random forest-based feature selection	Multimodal attention-based deep learning	MCI, AD, and controls with 96.88% accuracy
Lee et al. (2020) [16]	Gene expression: ANM1, ANM2, ADNI	VAE, TF genes.	LR, L1-LR, SVM, RF, and DNN	AUC: 0.657, 0.874, and 0.804 for ADNI, ANMI and ANM2
Park et al. (2020) [17]	Gene expression: GSE33000, GSE44770, methylation data: GSE80970	Integrate DEGs and DMPs by inter-section	DNN (Deep Neural Network)	0.823 is the average accuracy
Kalkan et al. (2022) [18]	Gene expression: GSE63060, GSE63061, GSE140829	LASSO regression	CNN on transformed image representation	AUC of 0.875 for the AD vs. CTL. AUC of 0.664 for the MCI vs. AD. AUC of 0.619 for the MCI vs. CTL
Eyigoz et al. (2020) [19]	Linguistic biomarkers- derived from cookie theft picture analysis	*p*-value based feature selection, *p* value < 0.05	Logistic regression	AUC 0.74 and accuracy 0.70
R’mani Haulcy et al. (2021) [20]	Linguistic biomarkers- audio and text features	PCA	LDA, DT, KNN, SVM, RF	Accuracy 0.854 and 0.657 for model with text and audio feature, respectively
Fathi et al. (2024) [21]	MRI images from ADNI	Resizing, removing non-brain image slices	CNN with majority-voting and probability-based ensemble methods	Accuracy of 98.57, 96.37, 94.22, 99.83, 93.88, and 93.92 for NC/AD, NC/EMCI, EMCI/LMCI, LMCI/AD, 4-way and 3-way classification, respectively

**Table 2 diagnostics-14-02640-t002:** Recommended methods for AD diagnosis.

Biomarker Data Sets	Feature Selection/Feature Reduction	Data Imbalance with Low Sample Size	Suggested Supervised Learning Classifiers
Image (MRI/CT)	Local patch selection, filters (of CNN), autoencoder family of algorithms	Use pretrained model, use F1 score, AUC for performance, cross-validation	CNN, attention-based model, pretrained model, XGBoost, ensemble with CNN as fusion
Non-image clinical	Algorithms like SFS, correlation matrix	Increase weight of minority classes Use F1 score, AUC for performance, cross-validation	DL, SVM, RF, KNN, LR, XGBoost
Audio features: i-vectors and x-vectors	PCA	Use F1 score, AUC for performance, cross-validation	SVM, RF, NN, DT, ensemble
Text features: word vectors, BERT embeddings, LIWC, CLAN	PCA	Use F1 score, AUC for performance, cross-validation	SVM, RF, NN, DT LLM for transcript analysis
Genome expression, epigenetics data	Feature ranking algorithm, algorithm like SFS, autoencoding family of algorithms, selection of gene and epigenetic data	Merge different datasets of similar attributes/gene transcripts, increase weight of minority classes augment train data with synthetic data, use F1 score, AUC for performance, cross-validation	DL, CNN, XGBoost SVM (when sample size is low)

**Table 3 diagnostics-14-02640-t003:** List of biomarkers and their utility in AD diagnosis.

Biomarker	Collection Procedure	Early Diagnosis Scope	Diagnosis Accuracy
CSF fluid	Invasive	Medium—High	Medium—High
PET scan	Non-invasive with radioactive risk involved	High	Medium—High
MRI	Non-invasive	Medium	High
DTI	Non-invasive	Medium—Low	Medium
Blood plasma	Minimal invasive	Medium	High
MMSE	Non-invasive	Medium	Medium
APOE4 count	Non-invasive	High, preclinical	Medium—Low
Gene expression	Minimally invasive	High, preclinical	Medium
Gene sequencing	Minimally invasive	High, preclinical	Medium—Low
Speech and text	Non-invasive	High	Medium—Low
Demographic	Non-invasive	Medium—Low	Medium—Low
Combinational study	Invasive, non-invasive	High	High

## Data Availability

Not applicable.

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
