# Peer review of "Etiology of Late-Onset Alzheimer’s Disease, Biomarker Efficacy, and the Role of Machine Learning in Stage Diagnosis"

_diagnostics, 2024, doi:10.3390/diagnostics14232640_

Round 1

Reviewer 1 Report

Comments and Suggestions for Authors

Although the proposed manuscript has merits, there are some issues that need to be addressed.

Comment 1: Abstract should be slightly reformulated to explicitly define its purpose, such as evaluating the role of ML and biomarkers in diagnosing late-onset Alzheimer's disease (LOAD).  Abstract briefly mentions a review of research papers, but it lacks detail on how the studies were selected and what types of analysis were performed. Abstract should emphasize the results of the review. The author should highlight its contributions more clearly. While it lists gaps and discusses future possibilities like the use of Generative AI and SMOTE for data augmentation, it doesn't explicitly mention the contributions of its review to the existing body of knowledge. The conclusive section of the abstract needs to be much stronger. It should explicitly mention whether ML methods, biomarkers, or advanced data augmentation techniques can enhance the early diagnosis of LOAD and what challenges still need to be addressed in future research.

Comment 2: Include and clearly state research questions and contributions of proposed study in the introduction. Also include some kind of existing review similar to your review and bring out what they have not been able to achieve in their review that this present review is able to achieve.

Comment 3: The imaging modalities mentioned in the review are not limited to just DTI and MRI. It will be beneficial if other imaging modalities are also included in the review. Review should also include a novel and perspective research field – Multimodal neuroimaging. Several research have combined imaging modalities together as biomarkers for AD diagnosis. There are many multimodal applications in this area.

Comment 4: Describe all tables before it is presented e.g Table 1 represents the lists of biomarkers

Comment 5: Change the acronyms Ad to AD in section 4

Comment 6: The acronyms PCA, SVM, ANN, SPECT, are not defined before usage. This should be corrected ‘a DL (deep learning) after DL is defined. In general, work on all the acronyms used in the entire manuscripts including the abstract section. Some acronyms appeared in the abstract without being defined.

Comment 7: There are several words that the accronyms has been defined but the words have still been repeated in the entire manuscript. For instance: to classify Alzheimer disease (AD) genes.

Comment 8: Revise all tables and figures to align on the center of the page.

Comment 9: Contrast your findings with related survey.

Comment 10: Conclusion should be extended to include more details regarding the future work and limitations of proposed study.

Reviewer 2 Report

Comments and Suggestions for Authors

The topic of the paper is Late Onset Alzheimer’s Disease (LOAD), a specific pathological condition that manifests after 65 years. The goal is to explore potential biomarkers for early LOAD diagnosis through Machine Learning and Deep Learning models based on datasets and clinical instrumental examinations.

The paper presents multiple critical issues and gaps.

First, as it is a review, the authors should have explicitly clarified the criteria for selecting the papers included (reference period, selection keywords, electronic databases used).

Furthermore, the number of papers selected and analysed in a review is expected to be much higher.

In addition, the 'Deep Learning' aspect, which the title indicates as a central topic of the review, is the least explored in the paper (Section 4 mentions only [27]).

Then, Tables 1 and 2 are not referenced in the paper and the authors need to clarify how tables were populated. For example, in Table 1, how was the level of utility of each approach determined? What metrics were used to categorise as low-medium-high according to the approach, scope, and accuracy? Regarding Table 2, what are the sources of information included in the table (papers? Which ones?)

The Related Work section should be presented better. Generally, the primary information about the works included in a review is presented in tabular form to facilitate a quick comparison between the works and provide an immediate focus on the key details of the selected studies. This preliminary analysis would have also allowed the authors to organize their paper better, as it appears confusing and difficult to read.

The Results section is missing, as well as a Discussion with summary information on the selected studies and their application to the topic (LOAD) to provide an overview of the evidence and insights, potentialities and limitations of current approaches, and indications for future improvements.

The conclusions are weak and need to be extended according to the results of the analysis.

The primary focus on LOAD does not emerge from the paper: general approaches regarding AD were considered, but not specifically on Late Onset.

For all these reasons, the paper cannot be accepted for publication. It requires a thorough revision of the structure (which is currently ineffective), the content (which needs to be expanded), the presentation (which needs to be improved and detailed), the included works (which should extend and be more focused, especially regarding LOAD and Deep Learning), and finally the English language (the paper is currently not fluent and hard to read).

Comments on the Quality of English Language

The paper needs an extended revision of English, as it is hard to read. 

Round 2

Reviewer 2 Report

Comments and Suggestions for Authors

The authors have revised the paper in an effort to address the comments from the previous review round.

However, in my opinion, the changes made are neither sufficient nor effective in filling the gaps previously highlighted regarding clarity, readability, organization and number (and criteria) of selected papers. These features are crucial for a review article to provide an exhaustive overview of the state of the art.

For this reason, I cannot change my previous recommendation.

Additionally, I suggest that the authors revise the title before a future submission of their work. As previously noted, the term 'Deep Learning' in the title is misleading, as much of the paper discusses Machine Learning. Since Deep Learning is not a central topic but a result of their work, as indicated by the authors, I recommend including 'Machine Learning' in the title and keywords (e.g., '... : the role of Machine Learning and Deep Learning').   

Comments on the Quality of English Language

The quality of English has improved, and no critical issues were found in the revised version.

Round 3

Reviewer 2 Report

Comments and Suggestions for Authors

The authors have further revised the paper in terms of both organization and content. As suggested, they changed the title, which, in my opinion, better captures the key topics of the Review and provides greater coherence between the title, keywords, and content. Mentioning the use of Deep Learning among the results of the literature analysis is appropriate. The paper has improved in overall quality compared to previous versions.

Some minor changes:

  • It is preferable to avoid references in the abstract. Please move reference [1] to the Introduction.
  • Include full names for acronyms used in the paper; some acronyms (e.g., ADNI, SNUBH, etc.) are missing their full names.
  • Paragraph 3.3 “Efficacy of AD biomarkers” should be renumbered as 4.3.
  • The numbering of paragraphs 4 and 5 should be adjusted to 5 and 6.

Comments on the Quality of English Language

Some of the newly added sentences have some issues with English structure.
